# Phosphatidylserine: The Unique Dual-Role Biomarker for Cancer Imaging and Therapy

**DOI:** 10.3390/cancers14102536

**Published:** 2022-05-21

**Authors:** Ahmet Kaynak, Harold W. Davis, Andrei B. Kogan, Jing-Huei Lee, Daria A. Narmoneva, Xiaoyang Qi

**Affiliations:** 1Department of Biomedical Engineering, University of Cincinnati, Cincinnati, OH 45221, USA; kaynakat@mail.uc.edu (A.K.); leej8@ucmail.uc.edu (J.-H.L.); narmond@ucmail.uc.edu (D.A.N.); 2Division of Hematology and Oncology, Department of Internal Medicine, University of Cincinnati College of Medicine, Cincinnati, OH 45267, USA; harold.davis@ohiohealth.com; 3Physics Department, University of Cincinnati, Cincinnati, OH 45221, USA; koganab@ucmail.uc.edu

**Keywords:** cancer biomarkers, phosphatidylserine, saposin C, dioleoylphosphatidylserine, SapC-DOPS, electric field, cancer imaging, enhanced cancer therapy

## Abstract

**Simple Summary:**

Chemotherapy, radiotherapy and surgery are the primary therapies for cancer. Even with these current treatment modalities, the death rate for many cancers is still high. Thus, there is still an urgent need for new therapeutic approaches which are safer and more effective. Cancer biomarker targeting is a promising strategy for cancer treatment. Cancer cells are distinguished from normal cells by their unregulated differentiation and over or under-expression of certain biomarkers or alteration of genetic material. In this review, we discuss phosphatidylserine biomarker-targeted therapy and imaging modalities in pre-clinical and clinical studies. We also appraise chemotherapy, radiotherapy and electric field-enhanced biomarker-driven therapeutic approaches.

**Abstract:**

Cancer is among the leading causes of death worldwide. In recent years, many cancer-associated biomarkers have been identified that are used for cancer diagnosis, prognosis, screening, and early detection, as well as for predicting and monitoring carcinogenesis and therapeutic effectiveness. Phosphatidylserine (PS) is a negatively charged phospholipid which is predominantly located in the inner leaflet of the cell membrane. In many cancer cells, PS externalizes to the outer cell membrane, a process regulated by calcium-dependent flippases and scramblases. Saposin C coupled with dioleoylphosphatidylserine (SapC-DOPS) nanovesicle (BXQ-350) and bavituximab, (Tarvacin, human–mouse chimeric monoclonal antibodies) are cell surface PS-targeting drugs being tested in clinical trial for treating a variety of cancers. Additionally, a number of other PS-selective agents have been used to trigger cytotoxicity in tumor-associated endothelial cells or cancer cells in pre-clinical studies. Recent studies have demonstrated that upregulation of surface PS exposure by chemodrugs, radiation, and external electric fields can be used as a novel approach to sensitize cancer cells to PS-targeting anticancer drugs. The objectives of this review are to provide an overview of a unique dual-role of PS as a biomarker/target for cancer imaging and therapy, and to discuss PS-based anticancer strategies that are currently under active development.

## 1. Introduction: Biomarkers in Cancer Imaging and Therapy

Cancer is expected to rank as the leading cause of death worldwide in the 21st century [1]. Cancer can affect anyone regardless of sex, age or social status. In 2020, there were an estimated 1.8 million new cancer cases diagnosed and 606,520 cancer deaths [2]. In addition to the pain and suffering it causes, cancer can substantially diminish the patient’s ability to maintain a normal lifestyle, often requiring prolonged periods of hospitalization and informal care. These demands increase social and financial pressures on governments, institutions and families.

Chemotherapy, radiotherapy and surgery are the primary therapeutics in the cancer clinic. Over the last decade, with a better understanding of the role of the immune system in cancer development and progression, immunotherapy has become a promising addition to the arsenal of cancer treatments [3,4,5]. Even with these current treatment modalities, the death rate for cancer remains high. Thus, there is an urgent need for new therapeutic approaches that result in better patient survival with fewer toxic side effects.

Imaging has an important role in personalized cancer medicine and is performed widely for the detection and characterization of cancer, such as evaluating the stage of the tumor, detecting disease recurrence, monitoring therapy progression, or post-therapy surveillance [6,7,8].

According to the National Cancer Institute (USA), a biomarker is “a biological molecule found in tissues or in the body fluids that sign for normal or abnormal process” [9]. Cancer biomarker-targeting strategies show considerable promise for cancer treatment. While conventional cancer treatments elicit significant off-target effects on normal, healthy cells, drugs directed against biomarkers can specifically home in on cancer cells with fewer off-target effects [10]. Cancer biomarkers are also important molecular signatures of the cell phenotype that help in detection of cancer, even at an early stage. The biomarkers can be proteins (EGFR [11] and HER2 [12]), nucleic acids (miR-2 [13], miR-155 [14], BRCA1 [15], DAPK1 [16], and MGMT [17]), lipids (phosphatidylserine (PS) [18]), glycoproteins (α-fetoketoprotein [11] and CA125 [11]) or carbohydrates (CA19-9 [11]).

This review specifically focuses on PS and its unique dual role as a diagnostic tool (e.g., a cancer biomarker) and as a cancer therapeutic target. PS has shown promise for the development of non-invasive imaging technologies to support diagnoses and evaluation of treatment efficacy for cancer [19], inflammatory [20,21] and cardiovascular disease [22,23]. Owing to its biological flexibility, PS is generally superior to other membrane lipids for imaging and therapeutic endeavors.

## 2. PS: Cellular Distribution and Roles

While only 3–10% of all cellular lipids [24], PS is found in both prokaryotic and eukaryotic cells and plays crucial roles in a variety of cellular activities including presynaptic neurotransmitter release, post-synaptic receptor activity, mitochondrial membrane integrity and stimulation of protein kinase C in memory generation [25,26,27]. Normal cells exhibit most of their PS on the cytosolic side of the cell membrane (Figure 1). PS transfer between inner and outer cell membranes is regulated by a group of ATPases, and amino phospholipid translocases (flippases) [28]. Exposure of PS is classically known as a marker of apoptotic cells, where PS acts as a “eat me” signal for PS receptors on immune cells and leads to clearance of the cells by macrophages [29]. PS also plays a role in immune modulation. It is known that PS exposure on cancer cells also leads to immunosuppression in the tumor microenvironment, where it increases the activity of natural killer and dendritic cells and shifts the polarization of tumor-associated macrophages (TAMs) into anti-inflammatory (M2) macrophages [30,31].

## 3. PS as a Cancer Biomarker

### 3.1. Heterogeneity in Surface PS Exposure on Membranes May Indicate a Susceptibility of Cancer Cells to Cancer Treatments at Different Stages

In contrast to the normal cells, cancer cells are unable to preserve PS asymmetry, leading to the surface exposure of PS on cell membranes [18,32] (Figure 1). The extent of PS exposure on the outer cell membrane differs significantly between different types of cancer cells. Interestingly, this variability is seen even within the same cancer type. For instance, while one subset of cells in a cell line demonstrates limited surface PS exposure, another subset of cells from the same cell line exposes high PS on cell membranes [18]. Cancer cells with low surface PS appear to be more sensitive to irradiation and chemotherapies such as gemcitabine (Gemzar)/nab-paclitaxel (Abraxane) [33,34]. On the other hand, cancer cells with higher surface PS are more sensitive to PS-targeting anticancer treatments, including saposin C embedded in a dioleoylphosphatidylserine nanovesicle (SapC-DOPS) [33,34,35]. PS is also a driver of cancer metastasis and immune escape [29]. Moreover, there is a positive correlation between surface PS exposure of tumors and their malignancy [36]. Therefore, the level of surface PS exposure on cancer cells may be important for diagnostic differentiation between different stages of the cancer and/or the susceptibility of the cancer to various treatments.

Monitoring cell surface PS is a compelling approach to quantitatively assess tumor growth and progression because the standard cancer treatments, including chemotherapy and radiotherapy, can increase surface PS on endothelial and stromal cells in the tumor. Successful PS-targeted cancer cell imaging has been demonstrated by using different carriers [37,38]. Antibodies that specifically target PS, such as bavituximab and PGN635 (fully human PS-specific monoclonal antibody) and PS-binding annexins have been used in pre-clinical studies as imaging agents, as summarized in Table 1.

Annexin V, a protein which binds PS, has been used for imaging of many cancers [42,43,44]. However, annexin V has a very short, 3 to 7 minute, half-life in blood, which further restricts its use for clinical imaging [45]. To address this, Zhao et al. [40] developed PGN635, a human monoclonal antibody against β2 glycoprotein 1 (β2GP1)-associated PS that has a longer blood half-life (~16 h). They labelled the F(ab’) (2) fragment of PGN635 with near-infrared dye, IRDye 800CW, and used it for optical imaging of U87 glioma xenografts in a mouse model. They successfully imaged the tumor 4 h post-IV injection of PGN-800 CW, with the highest signal observed at 24 h. The image signal was further enhanced by irradiation, where both PGN-800 CW tumor uptake and image contrast were pronounced once the tumor was irradiated with 6 Gy [40]. Similar results were obtained by Gong et al. following prostate tumor treatment with docetaxel [46].

Radiolabeled PS-targeted antibodies were also used to develop positron emission tomography (PET) imaging techniques. PET imaging was used to assess the efficacy of ^89^Zr-labelled PGN635 on tumor xenographs in mice. In those tumors, undergoing apoptosis, a strong accumulation of ^89^Zr-PGN635 was detected, attaining 30% ID per gram tissue with tumor-to-blood ratios of up to 13 [37]. In another study using the Dunning prostate carcinoma rat model, ^74^As-labeled bavituximab was effectively used to visualize the tumors [41]. In that study, PET imaging data showed that the tumor–liver ratio was 22 for bavituximab compared with 1.5 for the control, 72 h after injection [41].

### 3.2. SapC-DOPS Nanovesicles Can Target Cancer Surface PS

Different cancer cell lines can also be monitored using a system that includes PS and SapC, which is an endogenous sphingolipid activator protein that plays an important role in lysosomal enzyme activation and sphingosine and ceramide generation from sphingolipid degradation [47,48,49,50,51,52,53]. SapC has strong binding affinity for PS at an acidic pH [49,50,51,52,54,55]. It is known that tumors express abundant PS on the cell surface and have a lower extracellular pH (pH~6) than normal tissues (pH~7) due to lactate secretion from anaerobic glycolysis [56]. Therefore, the SapC-PS interaction provides a valuable, highly selective system for targeted tumor imaging and therapy. Previous studies have shown that nanovesicles comprised of SapC and DOPS, SapC-DOPS can selectively target tumor cells [24,34,47,57,58,59,60,61,62]. Our group has also shown that contrast agent-loaded SapC-DOPS nanocarriers can be used to monitor and trace different cancer cell lines [61,63]. Successful imaging of cancer cells with the PS-targeting SapC-DOPS nanocarrier system has been reported using optical, magnetic resonance imaging (MRI), as well as single-photon emission computed tomography (SPECT) [24,64,65], as described below. Figure 2 shows a schematic representation of SapC-based imaging modalities.

#### 3.2.1. Optical Cancer Imaging Using PS-Targeting SapC-DOPS Nanovesicles

Optical imaging has been widely used in pre-clinical cancer research [66,67]. In particular, it is utilized to conduct research on cancer markers, drug pharmacodynamics and to examine the effects of drugs in small animals [68]. The allure strength of optical imaging is that it is economical, easy to manage, and provides real-time results.

CellVue Maroon (CVM) is a far-red fluorophore and Chu et al. [69] demonstrated that CVM- tagged SapC-DOPS nanovesicles can be utilized for detecting brain tumors and arthritic joints in mice. A rotational bed was used to obtain the multi-angle rotational optical image. The results confirmed that optical imaging intensity depends on the optical imaging angle, which varies with cancer type in different animal models. For example, the values for the optical imaging angle in orthotopic and mut49 tumor-bearing mouse models were determined to be 10° and 20°, respectively. In the same study, Chu et al. showed that SapC-DOPS nanovesicle-based optical imaging not only provided information about the disease or the cancer site, it also enabled assessment of disease state and/or cancer progression [69]. Similar results have been reported by Kaimal et al. in their mouse xenograft models of pancreatic adenocarcinoma, neuroblastoma and a murine rhabdomyosarcoma model [64]. However, while optical imaging is highly sensitive, it has significant limitations. For example, optical imaging techniques have limited utility in human patients due to challenges associated with signal detection of fluorescence probes in deep tissues. Another limitation is autofluorescence that is generated in animal/human tissue and in ingested food. Studies are ongoing to circumvent these limitations; thus, Blanco et al. (see Section 3.2.3) demonstrated that phenol-substituted membrane-intercalating lipophilic dyes and conjugated iodinated lipophilic dyes can be incorporated into SapC-DOPS, allowing dual imaging of glioblastoma (optical and PET imaging) [61].

#### 3.2.2. Magnetic Resonance Imaging (MRI) Using PS-Targeting SapC-DOPS Nanovesicles

MRI is a broadly used method for tumor detection. Even though its sensitivity is low, MRI has exceptional soft tissue contrast and excellent spatial resolution [70]. MRI was used to selectively image neuroblastoma by utilizing iron oxide particles encapsulated inside SapC-DOPS nanocarriers [64]. By tagging these vesicles with ultrasmall superparamagnetic iron oxide contrast agent (USPIO) particles, Kaimal and his-coworkers monitored both in vitro and in vivo delivery and uptake of the SapC-DOPS-IO vesicles. According to the in vitro results, there is a significant increase on the R_2_ and R_2_^*^ relaxation rates (14.64 s^−^^1^, 26.74 s^−^^1^) once the cells are exposed to SapC-DOPS-IO for 24 h as compared to the control cells (7.84 s^−^^1^, 11.04 s^−^^1^). According to an in vivo study conducted in mice, T_2_*-weighted imaging at 7T shows that the signal intensity in tumors drops immediately after injection of SapC-DOPS-IO, followed by a gradual further decrease before rebounding slightly 24 h later. The drop in signal intensity is observed throughout the tumor. Inductively coupled plasma atomic emission spectroscopy (ICP-AES) analysis shows that the concentration of iron in the tumor of a mouse injected with SapC-DOPS-IO is approximately 5-fold higher than the concentration of iron in the tumor of a free IO-treated mouse [64].

Winter et al. used paramagnetic gadolinium chelates, gadolinium-DTPA-bis(stearylamide) (Gd-DTPA-BSA)-loaded SapC-DOPS vesicles as a targeted contrast agent for imaging glioblastoma multiform tumors [63]. At 7T, Gd-DTPA-BSA/SapC-DOPS vesicles and Gd-DTPA display a similar relaxivity of 3.32 and 2.80 (s·mM)^−1^, respectively. According to the in vivo experiments using injection of Gd-DTPA-BSA/SapC-DOPS vesicles, the R_1_ values before the injection of the tumor, contralateral normal brain, and sham-injected brain were 0.4676 ± 0.010 s^−^^1^, 0.5596 ± 0.003 s^−^^1^ and 0.5216 ± 0.034 s^−^^1^, respectively. Following the injection, the average change in the tumor R_1_ value was 9.0 ± 2.3% (*p* < 0.05) at 10 h post-injection, whereas the normal brain and the sham-injected brain showed no significant change, 1.2 ± 1.5% and 1.4 ± 1.9% (*p* > 0.05). The tumor R_1_ was increased (7.9 ± 1.5%, *p* < 0.05) compared to that for the normal and sham brains at 20 h post-injection, and it became statistically indistinguishable from the controls at 24 h post-injection (4.7 ± 2.0%, *p* > 0.05). Figure 3A shows the T_1_ maps of tumor cells before treatment and after 10 h injection of Gd-DTPA-BSA/SapC-DOPS vesicles. The results indicate that there is a clear reduction in the T_1_ relaxation time 10 h after treatment with Gd-DTPA-BSA/SapC-DOPS vesicles as compared with the T_1_ value before the injection. Figure 3B shows the percent change in T_1_ relaxation of sham tumor and sham normal brain treated with only SapC DOPS. At 4, 10, and 20 h, the increase in T_1_ relaxation time is higher in the sham normal brain, as compared with that in the sham tumor. By 24 h post-injection, the T_1_-weighted signal is similar in both tissues. Figure 3C shows the percent change in T_1_ relaxation after injection of Gd-DTPA-BSA/SapC-DOPS vesicle. At 4 and 10 h post-injection, the T_1_ relaxation time change is higher in the tumor (−4.12%, −4.05%) and the tumor rim (−3.81%, −4.94%), as compared with the T_1_ relaxation time change for the normal brain (−0.76%, −1.84%).

#### 3.2.3. Postron Emission Tomography/Single Photon Emission Computed Tomography (PET/SPECT) Imaging Using PS-Targeting SapC-DOPS Nanovesicles

PET and SPECT are imaging techniques commonly used in the clinic. They are used to detect gamma rays emitted from radioactive tracers given to the patients. The development of radiotracers has accelerated since both PET and SPECT are very sensitive and widely available.

A recent study by Blanco et al. used phenol-substituted membrane-intercalating lipophilic dyes labeled SapC-DOPS with iodine-127 for PET imaging [61]. The labeled SapC-DOPS colocalized with the bioluminescence signal in tumors and increased significantly after 1 h following the injection. Control experiments with iodine-125 conjugated to SapC-DOPS with the same phenol-substituted dye showed a 4- to 8-fold higher uptake in glioblastoma, as compared with a lower uptake in the sham brains, and a very low uptake in the thyroid. These results suggest that there is selective tumor targeting, and that there is only minimal reporter degradation in blood [61].

## 4. PS in Targeted Cancer Therapies

### 4.1. Therapies Using PS-Specific Targeting Agents

Because it has diverse biological roles [25,71], PS has attracted attention as a distinct therapeutic target among the other membrane lipids. The three major experimental PS-targeting agents used for cancer localization and treatment are SapC-DOPS [34,64,72,73], bavituximab, a monoclonal antibody that recognizes PS [74,75,76], PPS1D1, a PS-binding peptide–peptoid hybrid [77,78], and BPRDP056, a zinc (II)-dipicolylamine-SN38 conjugate [79,80]. A summary of PS-targeted therapies is presented in Table 2.

In addition to its applications in imaging as described in the Section 3, the SapC-DOPS nanodrug can also be used in anticancer therapy. At approximately 200 nm in diameter, the formulated SapC-DOPS nanovesicles have been demonstrated to selectively target and kill a variety of cancer cells including glioblastoma [90,91], pancreatic cancer [58,92], lung cancer [81], skin cancer [72], breast cancer [59], and pediatric tumors (neuroblastoma and peripheral nerve sheath tumor) [47]. SapC-DOPS induces cancer cell apoptosis [47,57,58,93] and lysosomal-mediated cell death [90]. SapC-DOPS also induces cytokine production in macrophages [94].

Bavituximab is a chimeric monoclonal antibody that binds to a complex of PS and β2GP1 to activate a T-cell-driven immune pathway and also blocks PS immunosuppressive signaling from tumor cells. Bavituximab binds Fc gamma receptors on myeloid-derived suppressor cells (MDSCs), M2 macrophages and immature dendritic cells, which leads to increased production of TNFα and IL-12 immunostimulatory cytokines. Consequently, it induces MDSC differentiation into M1-like macrophages and dendritic cells that cause the induction of tumor-specific cytotoxic T cells [82]. The bavituximab–paclitaxel combination has been used as a treatment for HER2-negative breast cancer in a phase I clinical trial. Treatment was well tolerated and resulted in an overall response of 85% [76]. Recently, bavituximab has been investigated in a phase III clinical trial for advanced stage lung cancer. Unfortunately, the combination of bavituximab with docetaxel does not enhance the efficacy comparing docetaxel alone in patients previously treated for non-small-cell lung cancer (NSCLC). Furthermore, the addition of bavituximab does not significantly change systemic adverse effects [74].

PS is specifically recognized by PPS1D1, a dimeric form of a peptide–peptoid hybrid. The monomeric form, PPS1, consists of distinct positively charged and hydrophobic residue-containing regions but is inactive. However, PPS1D1 displays strong cytotoxicity to lung cancer cells with no significant effect on normal cells in vitro, and it reduces tumor growth in vivo. Moreover, PPS1D1 significantly enhanced the efficacy of docetaxel in mice bearing H460 lung cancer xenografts [77,78].

Phosphatidylcholine–stearylamine (PC-SA) is a cationic liposome which specifically targets cancer cells. PC-SA induces apoptosis and shows potent anticancer effects as a single agent against most cancer cell lines. Additionally, in combination with doxorubicin (PC-SA-DOX), it results in a complete remission of B16F10 melanoma in C57BL/6 mice without signs of toxicity [84]. PC-SA-DOX also shows immunomodulatory activity by elevating Th1 cytokine levels in the tumor microenvironment, thereby facilitating treatment of lung metastasis [85].

Zinc (II) dipicolylamine-SN38 conjugate, BPRDP056 is a novel compound whose effect has been shown in pre-clinical studies with using variety of tumor models including colorectal, pancreas, prostate, liver, breast and glioblastoma. Because it contains zinc (II) dipicolylamine, BPRDP056 binds to cancer cell surface PS with strong affinity and SN38 induces apoptotic cell death in cancer. BPRDP056 exhibits significant activity in a dose-dependent manner and significantly inhibits the tumor growth compared to the controls [79,80].

Mch1N11 is a PS-targeting antibody. It has been shown that the combination of mch1N11 with the checkpoint inhibitors, anti-CTLA-4 or anti-PD-1 is superior to anti-CTLA-4 or anti-PD-1 alone in a melanoma mouse model. Furthermore, this combination increased the infiltration of CD4+ and CD8+ T cells into the tumor [89]. In another study, it was demonstrated that the combination of mch1N11 with anti-PD-1 significantly increased the antitumor activity with longer survival in a triple-negative breast cancer mouse model. The combination of mch1N11 with anti-PD-1 check point inhibitor also significantly elevates the number of tumor infiltrating lymphocytes and expression of pro-immune activating cytokines while downregulating the expression of pro-tumorigenic cytokines [88].

### 4.2. Application of Electric Fields to Enhance PS-Targeted Therapies

#### 4.2.1. General Considerations for Using Electric Field-Based Therapies

Electric fields and currents are helping fight cancer by destroying tumor cells without harming normal cells, both as a replacement of conventional treatments and as an adjuvant therapy [95,96]. Commonly, generating electric currents in a human or animal tissue by electric field is achieved using a pair of conductive electrodes between which a potential difference is maintained. When this potential difference is applied to cells or tissues, it results in heating, where the dissipated energy can be calculated based on the current (I), voltage (V) and duration of electric field application (t):Energy = (V) (I) (t)

There are several important considerations that need to be considered to ensure safety and efficacy of electric field therapy. It is critical to keep the dissipated energy as low as possible to avoid thermal effects on the tissues. Additionally, electrodes that are implanted into specific areas of the body produce an electric field that could essentially destroy all the cells in the vicinity of the target tissue, including both the tumor cells and normal cells. Correct placement of electrodes requires knowledge of the dielectric properties of the various tissue types and appropriate positioning of the conductors in the setup. Other necessary biological properties of note when applying an electric field to living organisms are: cell cycle phase, which affects the geometrical characteristics of the cells; extracellular environment (electrolytes in the interstitial fluid) that determines the molecular charge and the cell transmembrane potential; and ionic concentrations in the intracellular compartments, because ions such as Ca^2+^, Na^+^ and K^+^ carry the electrical current within the cells [97]. Below, we will review two common electric field-based treatment modalities used in the clinic to treat various cancers, e.g., tumor-treating fields (TTFields) and electroporation.

#### 4.2.2. Tumor-Treating Fields

TTFields, an intermediate frequency (100–300 kHz) alternating electric field, were approved by the FDA in 2011 for the treatment of adults with recurrent or newly diagnosed glioblastoma multiforme (GBM) [98,99]. For these patients, TTFields devices are applied to the scalp to deliver intermediate frequency alternating electric fields of low intensity. The devises consist of four transducer arrays, each consisting of nine insulated electrodes. Table 3 shows TTFields parameters that have been tested in a variety of cancer cell lines.

In 2004, Kirson et al. [101] showed that the inhibitory effect of TTFields is focused on proliferative cells, while quiescent cells are not affected. The mechanism of action is not fully understood; however, one theory suggests that TTFields (1–4 V/cm, 200 kHz, 24 h duration) act on rapidly dividing cells during metaphase, anaphase and telophase of mitotic cell division. At this frequency, alternating external fields induce inhomogeneous internal fields at the bridge between daughter cells, resulting in unidirectional forces that induce dielectrophoresis, interfere with the orientation of tubulin in the mitotic spindle, and cause mitotic catastrophe and mitotic cell death. The first clinical study that utilized TTFields was reported in 2004 with 20 GBM patients. The treatment modality was well tolerated and did not cause significant toxicity. The subsequent EF-11 phase III study showed that although there was no improvement in the overall survival vs. standard of care chemotherapy, the efficacy and activity of chemotherapy-free TTFields was comparable to chemotherapy, and the toxicity and quality of life clearly favored the TTField treatment [102].

Overall, a combination of TTFields treatment with conventional chemotherapy or radiation therapy is very promising in terms of prolonging the survival of patients as well as resulting in fewer adverse effects, as compared to the traditional methods alone. So far, the only adverse effect that has been reported is skin irritation under the transducer array [103]. On the other hand, the potential benefits of improved survival and reduced side effects are relatively modest considering the very large increase in treatment cost of approximately 200,000 USD per patient in the USA [104].

#### 4.2.3. Electroporation

Electroporation (or electropermeabilization) is a phenomenon which increases cell membrane permeability via inducing an electric field across the cell membrane. This permeabilization can be transient (reversible electroporation) or persistent (irreversible electroporation), depending on the electrical field parameters such as magnitude, exposure time, number of pulses [105]. Reversible electroporation is generally used in vitro to help the penetration of macromolecules, which are not capable of passing across the cell membrane by themselves.

Over the last three decades, irreversible electroporation (IRE) has been used to permanently permeabilize the cell membrane. The mechanism of electroporation has not yet been elucidated. One mechanism is that the electric field polarizes the cell membrane by changing the transmembrane potential across the cell membrane. The unstable membrane’s shape is altered, forming nanoscale pores through the membrane [106,107]. Electroporation-based cancer treatments are commonly used in the clinic and were first used as a treatment of cancer in the late 1980s in the form of electrochemotherapy (the use of electroporation to transfer non-permeant chemotherapy drugs into cells) [108]. Many in vitro or in vivo studies have shown that electrochemotherapy is a powerful, safe, low-cost treatment without significant adverse effects. Electrochemotherapy has been used in the treatment of many cancers, including melanoma, sarcoma, metastatic breast cancer and skin cancers [109,110,111,112]. Calcium electroporation is a novel anticancer treatment that has been investigated pre-clinically and in clinical trials, showing promising outcomes [113,114].

### 4.3. Surface PS Modulation via Electric Fields Is a Novel Approach to Increase Efficacies of Anticancer Therapies

Cancer cells that have high surface PS are more sensitive to PS-targeted treatments. Our data also suggest that cancer cells that have low PS exposure are more sensitive to radiation [34] or chemotherapeutics such as gemcitabine [33]. An important implication of these results is that altering the level of PS exposure in cancer cells is a logical approach to sensitize cancer cells to PS-targeting drugs such as SapC-DOPS and bavituximab. Wojton et al. [90] have shown that the combination of SapC-DOPS and temozolomide (Temodar) in glioblastoma displays a strong synergistic effect, as compared with the therapeutic effects of temozolomide alone. The potential mechanism may be that induction of apoptosis by temozolomide increases tumor PS exposure, thereby sensitizing GBM cells to the cytotoxic actions of SapC-DOPS [90].

PS externalization is regulated by increased intracellular calcium, which inhibits the activity of flippases [18]. Induced electric fields (EF) in the cell membrane modulate cell membrane potential [115], which alters the activity of calcium channels.

We have recently showed that EF can be used to modulate surface PS [116,117]. In these experiments, a direct current (DC)-EF was applied to GBM cells via a parallel-plate capacitor device with no direct contact between the cells or cell medium and electrodes (Figure 4). Our data showed that application of DC-EFs of varying magnitudes (75–150 V/cm) resulted in significant changes in the cell surface PS level in U87ΔEGFR-Luc glioblastoma cells (Figure 5) without effecting the total cellular PS levels [116,117]. In contrast to high-power or direct-contact EF applications, this low-amplitude, non-contact electric field modality does not affect the viability of normal, untransformed cells, and therefore can be expected to have minimal side effects, in contrast to those of cytotoxic chemotherapies or radiation. These data suggest that external non-contact DC-EFs might be a novel platform for enhancement of GBM treatment efficacy.

## 5. Conclusions and Future Directions

The important functional property of PS as a unique yet ubiquitous cancer cell biomarker makes it an appealing target for the development of novel anticancer therapies that could potentially be quick to translate into clinical applications. While many specific biomarkers have been reported for different cancer cell types, the ubiquitous nature of PS allows its application to target most cancers. Currently, there are several ongoing clinical trials that aim to achieve better therapeutic outcomes, including the trials for SapC-DOPS and bavituximab. Similarly, the unique properties of PS provide important advantages in imaging applications, where PS-targeting modalities enable us to selectively image tumors and will undoubtedly have a major role in future diagnostics applications.

In summary, emerging technologies that enhance PS-targeting treatment and imaging efficacy have tremendous potential for future cancer diagnostics and treatment. The analysis of the current literature also suggests that the highest therapeutic impact will likely result from the combination of PS-targeting therapies with already approved therapies. For example, altering surface PS exposure on cancer cells with electric fields may enhance the efficacy of PS-targeting drugs to increase tumor cell death, while reducing the dose of the more cytotoxic drugs. In the future, this novel type of EF field treatment must be studied in vivo and in the clinic with different types of tumors. PS-targeting therapies can be combined with chemotherapy [86], radiation [87], and immune checkpoint inhibitors (including antibodies targeting CTLA-4, PD-1, and PD-L1) [88,89] to significantly improve the outcome.

## Figures and Tables

**Figure 1 cancers-14-02536-f001:**
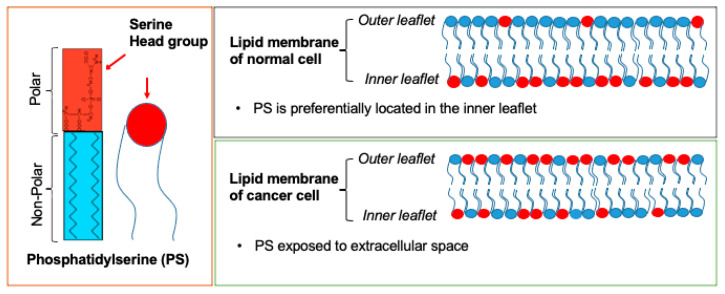
Schematic representation of phosphatidylserine (PS) and its distribution on normal and cancer cells.

**Figure 2 cancers-14-02536-f002:**
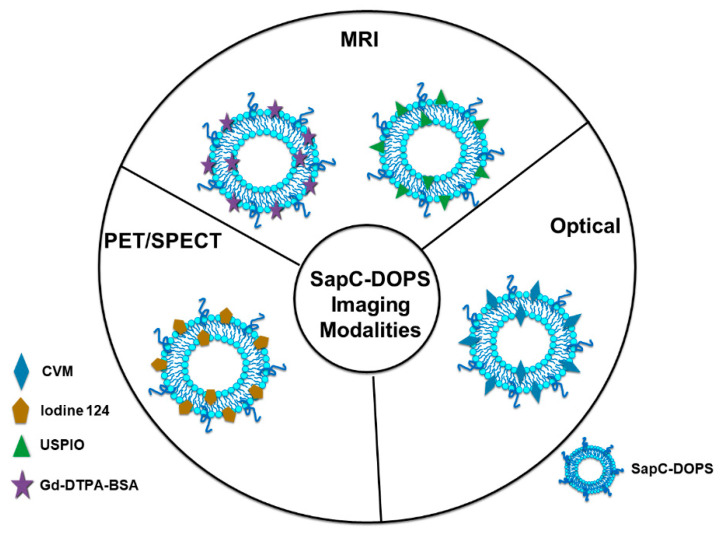
Schematic representation of saposin C-dioleoylphosphatidylserine (SapC-DOPS)-based tumor imaging modalities. For in vivo and in vitro studies, SapC-DOPS nanovesicles can be labeled with far-red fluorophore, CellVue Maroon (CVM) for optical imaging. For in vivo magnetic resonance imaging (MRI) imaging, the gadolinium chelates, gadolinium-DTPA-bis(stearylamide) (Gd-DTAP-BSA) or the ultrasmall superparamagnetic iron oxide (USPIO) can be incorporated and used as MRI contrast agents. For in vivo positron emission tomography (PET)/single-photon emission computed tomography (SPECT) imaging, SapC-DOPS can be combined with iodine-124 contrast agent.

**Figure 3 cancers-14-02536-f003:**
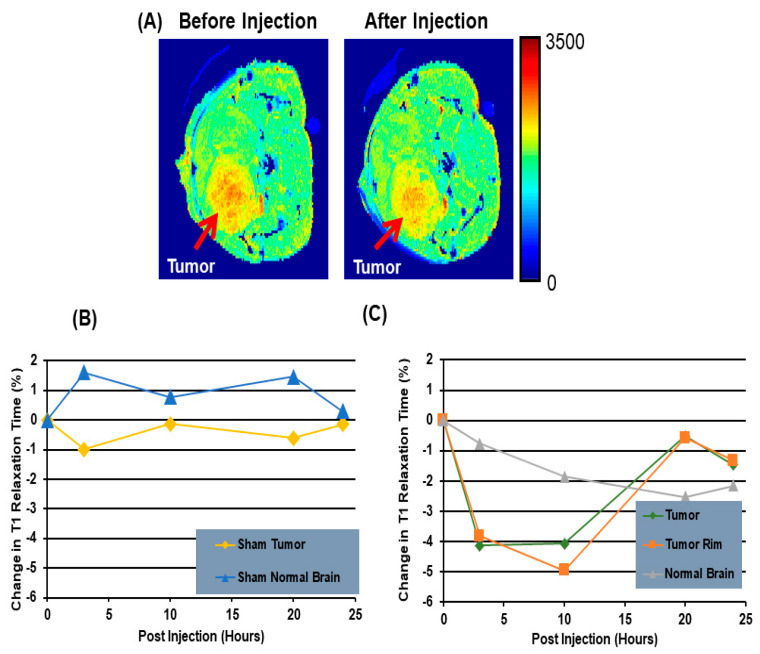
Use of saposin C-dioleoylphosphatidylserine (SapC-DOPS) as a carrier for magnetic resonance imaging (MRI) contrast agents in a mouse brain cancer model. (**A**) High resolution MRI of a glioma in a mouse. Tumor T1 relaxation time (s^−^^1^) maps before and 10 h after injection of gadolinium-DTPA-bis(stearylamide) (Gd-DTPA-BSA)/SapC-DOPS vesicles. (**B**) Percent change in T1 after only SapC-DOPS vesicle injection in the sham tumor and sham normal brain. (**C**) Percent change in T1 after injection of Gd-DTPA-BSA/SapC-DOPS vesicle in the tumor, tumor rim cells and normal brain.

**Figure 4 cancers-14-02536-f004:**
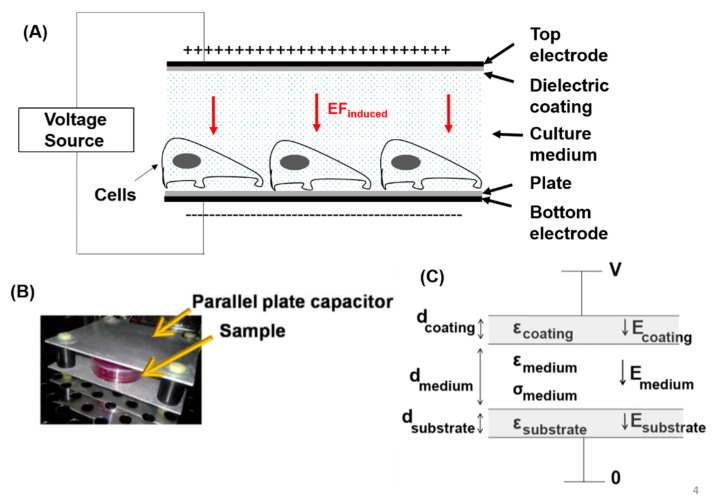
A custom system for direct current electric field (DC-EF) electrical stimulation of cells using the low-amplitude electric field and capacitive coupling method. (**A**) Schematic of the setup, where a cell culture plate is placed inside a parallel plate capacitor, and the electric field is perpendicular to the substrate thus preventing directional cell migration. (**B**) A photograph showing the capacitor with the cell culture plate inside. (**C**) Schematic of the boundary conditions and electrical properties of the system.

**Figure 5 cancers-14-02536-f005:**
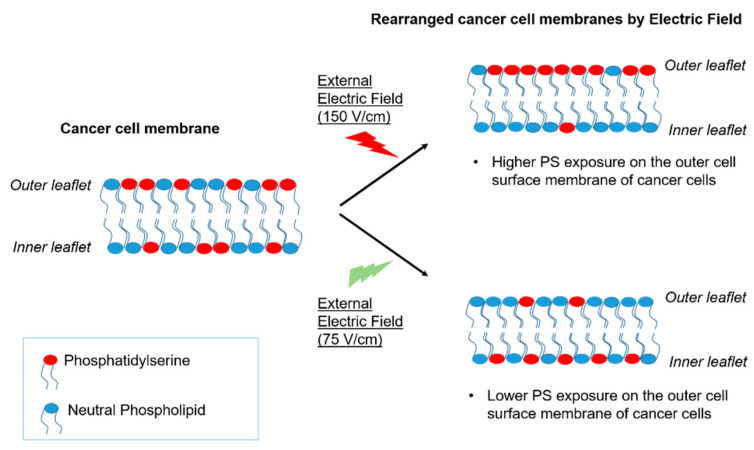
Schematic representation of phosphatidylserine (PS) modulation in cancer cells via external electric field.

**Table 1 cancers-14-02536-t001:** Phosphatidylserine (PS)-targeting imaging modalities.

PS-Targeting Imaging				
Type of Imaging Modality	PS-Targeting Antibody + Imaging Compound	Results	Cancer Type(s)	Refs.
Optical Imaging	Annexin V-Cy	A 2- to 3-fold stronger near-infrared fluorescent signal was observed in tumors of mice once treated with pro-apoptotic drug, cyclophosphamide.	Gliosarcoma	[39]
Optical Imaging	PGN635+ 800CW	Successfully imaged the tumor 4 h post-IV injection of PGN635-800 CW. Highest signal observed at 24 h.	Glioblastoma	[40]
Positron Emission Tomography (PET)	PGN635 + ^89^Zr	High accumulation ^89^Zr-PGN635 in treated tumors undergoing apoptosis. Tumor-to-blood ratios of up to 13.	Human colorectal cancer Breast cancer	[37]
PET	^74^As-labeled bavituximab	Tumor–liver ratio was 22 for bavituximab compared with 1.5 for an isotype matched control chimeric antibody 72 h after injection.	Prostate cancer	[41]
Magnetic resonance imaging (MRI) (9.4T)	PGN635 + Superparamagnetic iron oxide nanoparticles (SPIO)	T2-weighted MRI detected a drastic reduction in signal intensity and T2 values of tumors at 24 h.	Breast cancer	[38]

**Table 2 cancers-14-02536-t002:** Phosphatidylserine (PS)-targeted therapy modalities.

**PS-Targeted Therapy**				
**Type**	**PS-Targeting Drugs or Antibodies**	**Mechanism**	**Cancer Type(s)**	**Refs.**
Proteoliposomal nanovesicles	Saposin C-dioleoylphosphatidylserine (SapC-DOPS) (phase I and II clinical trials)	Caspase-mediated apoptotic and lysosomal-mediated cell death	Brain, Colorectal, GI, Lung, Breast, Skin, Neuroblastoma, Breast Cancer	[47,58,59,61,72,81]
Monoclonal antibody	Bavituximab (phase III clinical trial)	T-cell-driven adaptive immune pathway activation through M1-TAMs	Prostate cancer Non-Small-Cell Lung Cancer	[74,82]
Peptide–peptoid hybrid	PPS1D1 (Pre-clinical)	Membrane disruption	Lung Cancer	[77,83]
Cationic liposomes	Phosphatidylcholine-stearylamine (Pre-clinical)	Caspase-mediated apoptosis	Melanoma Glioblastoma	[84,85]
Zinc (II) dipicolylamine-based conjugate	Zinc (II) dipicolylamine (Pre-clinical)	Caspase-mediated apoptosis	Colorectal, Pancreas, Prostate, Liver, Breast, Glioblastoma	[80]
**Combinational PS-Targeted Therapy**				
**Modality**	**PS-Targeting Antibody + Chemo/Radiation**	**Detailed Description**	**Cancer Type(s)**	**Refs.**
PS-targeting antibody + chemotherapy	3G4 + gemcitabine	Significant reduction in primary tumor growth and metastatic burden 14-fold increase in macrophage infiltration over controls	Pancreatic Cancer	[86]
PS-targeting antibody + radiation	2aG4 + radiation	Focal irradiation increased the percentage of tumor vessels with exposed PS from 4% to 26% 91% reduction in tumor vascularity was observed when 2aG4 was combined with radiation therapy Enhanced monocyte/macrophage infiltration into the tumor mass	Lung Cancer	[87]
PS-targeting antibody + immune activators and checkpoint inhibitors	mch1N11 + anti-PD-1 or anti CTLA-4	Elevated fraction of cells expressing proinflammatory cytokines including IL-2, IFN-γ, and TNFα, and increased the ratio of CD8^+^ T cells to MDSCs and Tregs in tumors	Breast Cancer Melanoma Tumors	[88,89]

**Table 3 cancers-14-02536-t003:** Tumor-treating fields (TTFields) parameters used to treat different cancer cells.

Variables			Tumor Type	Results	Refs.
Time/ Temperature	EF Intensity	Frequency			
72 h/ 18 °C	1.7 V/cm	(100–500 kHz)	F98 rat glioma cells	A significant reduction in cell viability was observed at all applied frequencies, with the maximal reduction at 200 kHz	[100]
72 h/ 18 °C, 24 °C, and 28 °C	1.0 and 1.7 V/cm	(100–500 kHz)	U-87 MG	The maximum reduction in cell viability was observed when the cells were treated with 1.7 V/cm (incubator temperature: 28 °C) at 200 kHz	[100]
72 h/ 18 °C	1.7 V/cm	(100–500 kHz)	A2780 human ovarian cancer cells	A significant reduction in cell viability was observed at all applied frequencies, with the maximal reduction at 200 kHz	[100]
72 h/ 18 °C, 24 °C, and 28 °C	1.3 and 1.7 V/cm	(100–500 kHz)	OVCAR-3 human ovarian cancer cells	The maximum reduction in cell viability was observed when the cells were treated with 1.7 V/cm (incubator temperature: 18 °C) at 200 kHz	[100]
24 h/ 34 °C	1 and 2.5 V/cm	(100–300 kHz)	B16F1 Mouse malignant melanoma	Maximum cell growth inhibition was observed at intensities of 1.35 V/cm with 120 kHz frequency	[101]
24 h/ 37 °C	0–3 V/cm	(100–500 kHz)	MDA-MB-231 Human breast carcinoma	Maximum cell growth inhibition was observed at intensities of 1.75 V/cm with 150 kHz frequency	[100]

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
