# Peer review of "Phosphatidylserine: The Unique Dual-Role Biomarker for Cancer Imaging and Therapy"

_cancers, 2022, doi:10.3390/cancers14102536_

Round 1

Reviewer 1 Report

Overall, this current manuscript provides a comprehensive review over the imaging modalities of Phosphatidylserine in the preclinical or clinical stage. From the basic PS biology to the targeting concept of PS as a cancer biomarker, authors detailed the design, guiding principles, and development of several imaging modalities.

In section 4 for the therapies using PS-specific targeting agents, the authors should be aware of the body of literature in zinc dipicolylamine as small organic based molecule that has been employed for PS targeting. If there are concerns for including such work, the authors should provide the discussion on the reason for omission. 

The citation of reference 9 is missing.

Reviewer 2 Report

This review titled “Phosphatidylserine: The Unique Dual-Role Biomarker for Cancer Imaging and Therapyis a well-focused study on the different applications of targeting PS biomarker on cancer c ells in diagnosis and therapy. However, some important points need to be addressed.

  1. The overall writing needs thorough plagiarism check, as many a times line seem to be picked up from published papers.
  2. The authors could focus a bit on the role of PS targeting resulting in an immunomodulatory activity Since PS is also know as an upper checkpoint molecule itself, focusing on this would strengthen the review.
  3. Further, in the therapeutic aspects, authors have mentioned PS targeting in along with checkpoint blockade in the summarised table, but that is not at all discussed in the review, even though the concluding paragraph too mentions this as the future of cancer therapy. So, some light should be brought on this topic and discussed well in the therapy part.
  4. The conclusions seem to be a summary of all the papers, but quite vague on the whole and lacks the perspective of the authors on the topic of PS as target in cancer. Should be more well drafted.

Reviewer 3 Report

Cancer biomarkers are important molecular signatures of the cell phenotype that help in cancer diagnosis. The biomarkers can be proteins, nucleic acids, glycoproteins, carbohydrate, and lipids such as phosphatidylserine. In the article entitled “Phosphatidylserine: The Unique Dual-Role Biomarker for Cancer Imaging and Therapy”, the authors reviewed the use of phosphatidylserine as a biomarker/target for cancer imaging and therapy, as well as phosphatidylserine -based anti-cancer strategies. The review article is undoubtedly interesting in the field of oncological molecular nuclear medicine. The article is original, well-written, and well-organized. Therefore, I recommend the publication of the manuscript as is.

Round 2

Reviewer 1 Report

The author has adequately addressed the concerns.